# Gene-vegetarianism interactions in calcium, estimated glomerular filtration rate, and testosterone identified in genome-wide analysis across 30 biomarkers

Michael Francis[1¤]*, Kenneth E. Westerman[2,3,4], Alisa K. Manning[2,3,4], Kaixiong Ye[1,5]*

1 Institute of Bioinformatics, University of Georgia, Athens, Georgia, United States of America, 2 Clinical and Translational Epidemiology Unit, Mongan Institute, Massachusetts General Hospital, Boston, Massachusetts, United States of America, 3 Department of Medicine, Harvard Medical School, Boston, Massachusetts, United States of America, 4 Programs in Metabolism and Medical and Population Genetics, Broad Institute of MIT and Harvard, Boston, Massachusetts, United States of America, 5 Department of Genetics, University of Georgia, Athens, Georgia, United States of America

¤ Current address: Michael Francis, Booz Allen Hamilton, McLean, Virginia, United States of America
* mikefrancisphd@gmail.com; kaixiong.ye@uga.edu

**Data Availability Statement:** Full and annotated code used in this analysis, gene-level summary statistics, and annotated Miami plots are publicly

## Abstract

We examined the associations of vegetarianism with metabolic biomarkers using traditional and genetic epidemiology. First, we addressed inconsistencies in self-reported vegetarianism among UK Biobank participants by utilizing data from two dietary surveys to find a cohort of strict European vegetarians (N = 2,312). Vegetarians were matched 1:4 with non-vegetarians for non-genetic association analyses, revealing significant effects of vegetarianism in 15 of 30 biomarkers. Cholesterol measures plus vitamin D were significantly lower in vegetarians, while triglycerides were higher. A genome-wide association study revealed no genome-wide significant (GWS; $5\times10^{-8}$) associations with vegetarian behavior. We performed genome-wide gene-vegetarianism interaction analyses for the biomarkers, and detected a GWS interaction impacting calcium at rs72952628 ($P = 4.47\times10^{-8}$). rs72952628 is in *MMAA*, a $B_{12}$ metabolic pathway gene; $B_{12}$ has major deficiency potential in vegetarians. Gene-based interaction tests revealed two significant genes, *RNF168* in testosterone ($P = 1.45\times10^{-6}$) and *DOCK4* in estimated glomerular filtration rate (eGFR) ($P = 6.76\times10^{-7}$), which have previously been associated with testicular and renal traits, respectively. These nutrigenetic findings indicate genotype can modify the associations between vegetarianism and health outcomes.

## Author summary

The popularity of vegetarianism continues to rise, particularly among those seeking to improve their overall health. However, vegetarianism studies have been susceptible to imprecise definitions of dietary intake, and to biased selection of health-conscious

available at https://michaelofrancis.github.io/VegetarianGDI/. Summary statistics for GWAS and GWIS are available at GWAS Catalog (https://www.ebi.ac.uk/gwas/). The corresponding accession numbers can be found in S10 Table.

**Funding:** Research reported in this publication was supported by Foundation for the National Institutes of Health (T32GM007103 to MF), National Institute of General Medical Sciences (R35GM143060 to KY), National Institute of Diabetes and Digestive and Kidney Diseases (K01 DK133637 to KEW), National Heart, Lung, and Blood Institute (R01 HL145025 to KEW). The content is solely the responsibility of the authors and does not necessarily represent the official views of the National Institutes of Health. The funders had no role in study design, data collection and analysis, decision to publish, or preparation of the manuscript.

**Competing interests:** The authors have declared that no competing interests exist.

participants. Moreover, previous studies have not considered the modifying effects of individual genetic background. Here, we use data from two separate dietary surveys in the UK Biobank to define participants who were most likely to have followed a strict vegetarian diet. Using this cohort in an epidemiological analysis without consideration of genetics, we estimated the effects of vegetarianism on 30 serum biomarkers for cancer, diabetes, and cardiovascular, bone, renal and liver diseases. We replicated established health benefits of vegetarianism in lowering cholesterol, and found evidence of potentially negative effects, such as raised triglycerides and lowered vitamin D. Furthermore, we performed genome-wide gene-vegetarianism interaction analysis for these 30 serum biomarkers. We identified three novel genetic factors that modify the effects of vegetarianism on calcium, testosterone, and estimated glomerular filtration rate (eGFR). These novel gene-vegetarian interactions are the first of their kind, and all located within biologically relevant genes. These findings can assist in the design of personalized nutrition recommendations and future clinical trials.

## Introduction

Vegetarianism is a superordinate term covering a variety of animal-restricted dietary practices, typically referring to lacto-ovo vegetarianism, which permits plant-based food, dairy, and eggs, and excludes meat, fish, and seafood [1]. Estimates indicate that in Western countries, adherence to plant-based diets has increased over the past decade [2–4]. This has been motivated by several factors, including health benefits, taste preferences, ethical concerns with slaughtering animals and factory farming, environmental concerns related to pollution and greenhouse gas emissions, and perceived moral accreditation [4,5]. It is now common for nutrition professionals to recommend vegetarianism to the public *en masse* [4,6,7].

Recent large meta-analyses have found health benefits associated with vegetarianism, including improved blood lipids, and reductions in body mass index (BMI), heart disease, type 2 diabetes, and certain cancers; though no significant differences between vegetarians and nonvegetarians have been found in all-cause mortality [1,8–10]. As the authors of these meta-analyses have pointed out, many vegetarian observational studies are confounded by information and selection biases [1,8,9]. One way to alleviate these biases is to leverage modern biobank-scale datasets, which allow for rigorous subsetting and verification of study participants.

Heterogeneous and imprecise questionnaire design in defining vegetarianism has been a recurring source of information bias in previous studies. Self-reported vegetarians are known to vary widely in their strictness of following a diet that contains no meat or fish [11]. Furthermore, there are issues of trustworthiness in dietary questionnaire response, particularly in the direction of over-reporting behaviors perceived as healthy [12,13]. Dietary survey reliability can be improved by integrating multiple assessments to define intake, as opposed to use of a single survey instance [14–16].

Vegetarians may also be more health conscious in general than omnivores, which introduces a selection bias sometimes called the "healthy user effect" [17]. When lifestyle factors adjacent to vegetarianism are not properly controlled for, it can lead to overestimating the effect of vegetarianism. One outstanding example of this bias in vegetarianism studies, specifically those conducted in the US, has been an over-generalization of results from Seventh Day Adventists (SDAs) [1,8,9,18,19], who in addition to vegetarianism, observe many healthy lifestyle practices, such as increased emphasis on exercise, and avoidance of all tobacco, drugs, and alcohol. Meta-analyses reveal that non-SDA vegetarians consistently show fewer health

benefits than SDAs [1,8,9]. Matching participants across relevant lifestyle factors can mitigate this issue to a greater extent than solely including these factors as model covariates [20].

In addition to the aforementioned biases, large-scale studies of vegetarianism have not properly accounted for genetic differences across individuals. Genetics and ancestry are known to play an important role in metabolic processes (i.e., nutrigenetics [21]) and thus may impact both dietary behavior, and the effect of diet on disease risk. With regard to behavior, heritable components have previously been associated with plant-eating preference [22]. Specific genetic variants have been significantly associated with quantitative measures of plant-eating [23,24]. A recent GWAS in UK Biobank Europeans, which used a single survey instance to classify vegetarians, found one variant at genome-wide significance impacting vegetarianism status (rs72884519; $P = 4.997 \times 10^{-8}$) [25], while a GWAS in a Japanese cohort found none [26]. To discover how the health impacts of vegetarianism can depend on the genetic background of an individual, we considered gene-environment interactions (G×E), which can produce synergistic (non-additive) effects on health outcomes that genetic or environmental factors alone would not [27]. Gene-diet interactions (GDIs) are a type of G×E in which diet is the environmental exposure; for example, GDIs have been shown to impact biomarker levels when considering exposures of overall dietary patterns [28]. Identifying GDIs is a key step towards the implementation of precision nutrition recommendations that are tailored to the needs of an individual [29].

This study consists of four parts. First, by utilizing two types of dietary surveys administered to UKB participants, we defined a cohort of strict vegetarians who were most likely to be practicing this diet at the time of the serum biomarker collection. Second, we estimated effects of vegetarianism in a matched sample of vegetarian and nonvegetarian Europeans, across 30 serum biomarkers associated with diabetes, cancer, cardiovascular, skeletal, renal, and liver diseases. Third, we performed a genome-wide association study (GWAS) to search for variants that may explain vegetarianism behavior on a genetic level. Finally, we performed a genome-wide interaction study (GWIS) of vegetarianism across 30 biomarkers, identifying significant gene-vegetarianism interactions on calcium, estimated glomerular filtration rate (eGFR), and testosterone. This study provides the first biobank-scale evidence that GDIs play a role in differential health outcomes among vegetarians.

## Results

### Identifying a strict sample of vegetarians

We searched the UK Biobank (UKB) to find a strict subset of participants that were most likely to be vegetarian (S1 Fig), prioritizing data points nearest to the time of initial assessment (IA) when blood samples were collected for biomarker measurement. Two separate surveys were part of UKB dietary data collection, one at the IA which was taken by all UKB participants (N = 502,413), and one in the 24-hour recall survey (24HR), which was administered after the IA in five waves or "instances", between April 2009 and June 2012 (N = 210,967 unique participants; Fig 1A). Participants were invited to take the 24HR between one and five times on a voluntary basis (Fig 1B).

We used four criteria to designate participants as vegetarian. Our first criterion was whether a participant indicated they routinely followed a vegetarian or vegan diet; this question was only asked on the 24HR. A total of 9,115 participants self-identified in at least one 24HR that they were either vegetarian or vegan (Table 1). Because the sample size of strict vegans was too small (n = 133; S1 Table), these participants were pooled together with vegetarians in all analyses.

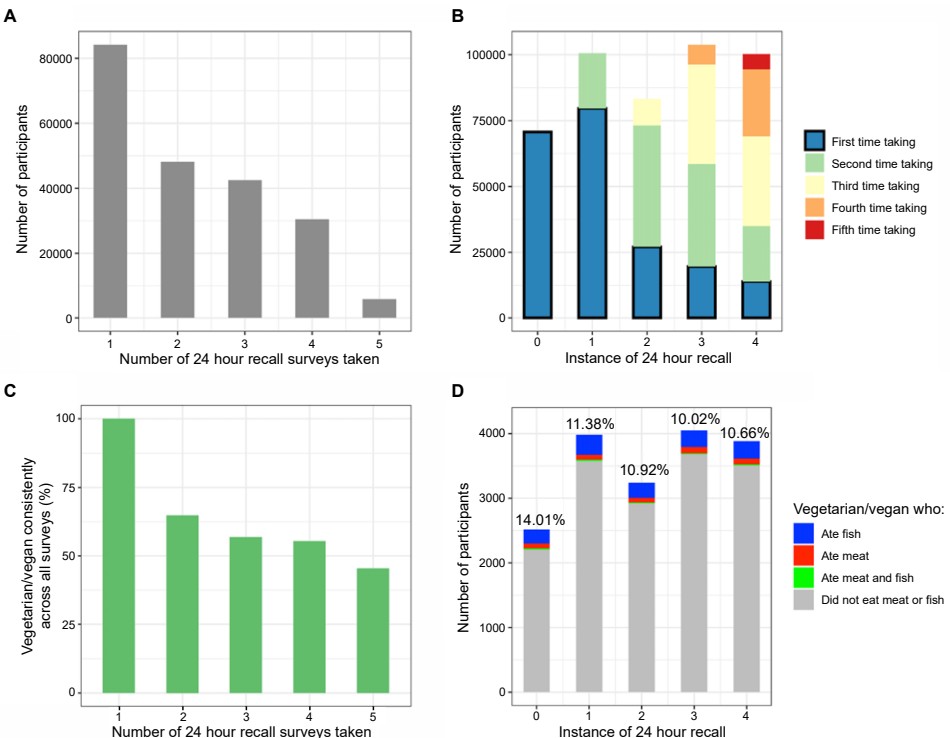

**Fig 1. Identifying vegetarians. (a)** Participants were invited to take the 24-hour recall survey (24HR) between one and five times on a voluntary basis. **(b)** The 24HRs we considered were restricted to the first time participants took that survey, because this was the closest time point to the blood draw of biomarkers at the initial assessment (Instance 0). **(c)** For participants who self-identified as vegetarian in at least one 24HR and took multiple 24HRs, they were less likely to self-identify as vegetarian in all surveys (over a 3 year period). **(d)** The percentage of vegetarians who indicated eating meat or fish on the same 24HR instance in which they identified as vegetarian ranged from 10.02–14.01% on all 24HR instances.

We found an inverse relationship between the percentage of participants who consistently self-identified as vegetarian in every 24HR they took, and the number of times participants took the 24HR (Fig 1C). For example, of the participants who identified as vegetarian at least once and participated in two instances of the 24HR, 64.8% self-identified as vegetarian both

**Table 1. Selecting strict vegetarians.** Vegetarians were filtered on four criteria: self-identifying as vegetarian on the first 24-hour recall survey (24HR) that they participated in, not eating meat or fish on the first 24HR, not eating meat or fish on the initial assessment (IA), and no major dietary changes over the past 5 years. Of 9,115 UK Biobank participants who self-identified as vegetarian at least once, 3,205 met these four criteria (top row, bold). The table shows counts from all UK Biobank participants who took the 24HR (N = 210,967); after filtering by ancestry, the total of 3,205 was reduced to 2,312 European vegetarians, the sample used in the analyses that follow.

| Self-reported vegetarian (first 24HR) | Ate meat or fish (first 24HR) | Ate meat or fish (IA) | Major dietary changes in past 5 years (IA) | N |
|---|---|---|---|---|
| **Yes** | **No** | **No** | **No** | **3,205** |
| Yes | No | No | Yes | 1,136 |
| Yes | Yes | No | No | 11 |
| Yes | Yes | No | Yes | 11 |
| Yes | No | Yes | No | 1,628 |
| Yes | No | Yes | Yes | 932 |
| Yes | Yes | Yes | No | 490 |
| Yes | Yes | Yes | Yes | 369 |
| Yes | No | N/A | N/A | 6 |
| No | - | - | - | 1,327 |

times (1,380 of 2,130); for participants that took the 24HR in all five instances, only 45.4% consistently identified as vegetarian every time (168 of 370). Because we were interested in biomarker levels at the IA time point, we considered self-identification as vegetarian in the earliest instance of the 24HR as sufficient for meeting this criterion.

Next, the 24HR asked whether a participant ate meat or fish yesterday. To find intra-survey discrepancies in vegetarianism status, we disqualified those who identified as vegetarian and also self-reported eating meat or fish on the same instance of the 24HR. The percentage of these participants ranged from 10 to 14% per survey instance (Fig 1D). Participants who reported eating meat on their first 24HR were disqualified from our strict vegetarianism status. Similarly, in our third criterion, we disqualified those who did not answer "Never" to questions about their frequency of eating meat or fish on the IA.

Lastly, because of the high amount of dietary fluctuation we found in self-identified vegetarians, we also required participants to have answered "No" to a question on the IA which asked whether they had any major dietary changes over the past five years. Overall, out of 9,115 UKB participants who self-identified as vegetarian (or vegan) on at least one 24HR, we found 3,205 met our criteria of not reporting eating meat on IA, nor on the nearest 24HR to the blood draw time point, and had not reported major dietary changes over the past five years (Table 1).

## Sample matching and estimating vegetarianism effects on serum biomarkers

After quality controlling participants and keeping only those who were part of the largest ancestry group, European, using Pan UKBB designations [30], 2,328 vegetarians and 153,047 nonvegetarians remained (S2 Table). Raw (untransformed) values for 30 traits were plotted, and some exhibited apparent differences between vegetarians and nonvegetarians (S2 Fig). However, the covariates selected for our effects estimation model (age, sex, BMI, alcohol use frequency, previous smoker, current smoker, Townsend index, and the first five genetic principal components) were highly imbalanced between the two groups (S2 Table). For example, the average ages of nonvegetarians and vegetarians were 56.5 and 52.7 years, respectively. Similarly, nonvegetarians were 54.1% female, compared to 66.2% in vegetarians. The covariates with highest standardized mean differences (SMD) between the two groups were age (-0.482) and BMI (-0.501). Therefore, prior to estimating the effects of vegetarianism across the 30 traits, we matched each vegetarian to four nonvegetarians along the covariates in the effect estimation model. After matching, the absolute SMD (ASMD) in all model covariates was <0.05 standard deviations (S3 Fig). The variance ratio of the distance of propensity scores between unmatched and matched vegetarians was improved from 2.12 to 1.02. Similarly, the maximum empirical cumulative density function (eCDF) difference (Kolmogorov-Smirnov statistic, $D_n$) was improved from 0.30 to 0.0013 (S3 Table). These measures indicate that covariate balance was achieved between matched vegetarians and nonvegetarians.

The standardized effect of vegetarianism was estimated across 30 serum biomarker traits with rank-based inverse normal transformation in 2,312 vegetarians and 9,248 matched nonvegetarians. Fifteen of these traits had significant effects at the Bonferroni-corrected $P$-value threshold of 0.05/30 = 0.0017, while five additional trait effects were nominally significant ($P<0.05$) (Fig 2 and S4 Table). Effects of vegetarianism were significant and negative across all cholesterol measures, including total cholesterol, low-density lipoprotein cholesterol (LDL), high-density lipoprotein cholesterol (HDL), plus Apolipoproteins A and B (ApoA, ApoB); while lipoprotein (a) (Lp (a)) was nominally significant. A significant positive effect of vegetarianism was observed on triglycerides ($\beta = 0.223$; $P = 4.0 \times 10^{-26}$). Vegetarianism had a significant negative effect on the steroid hormone vitamin D ($\beta = -0.388$; $P = 2.1 \times 10^{-49}$), and with

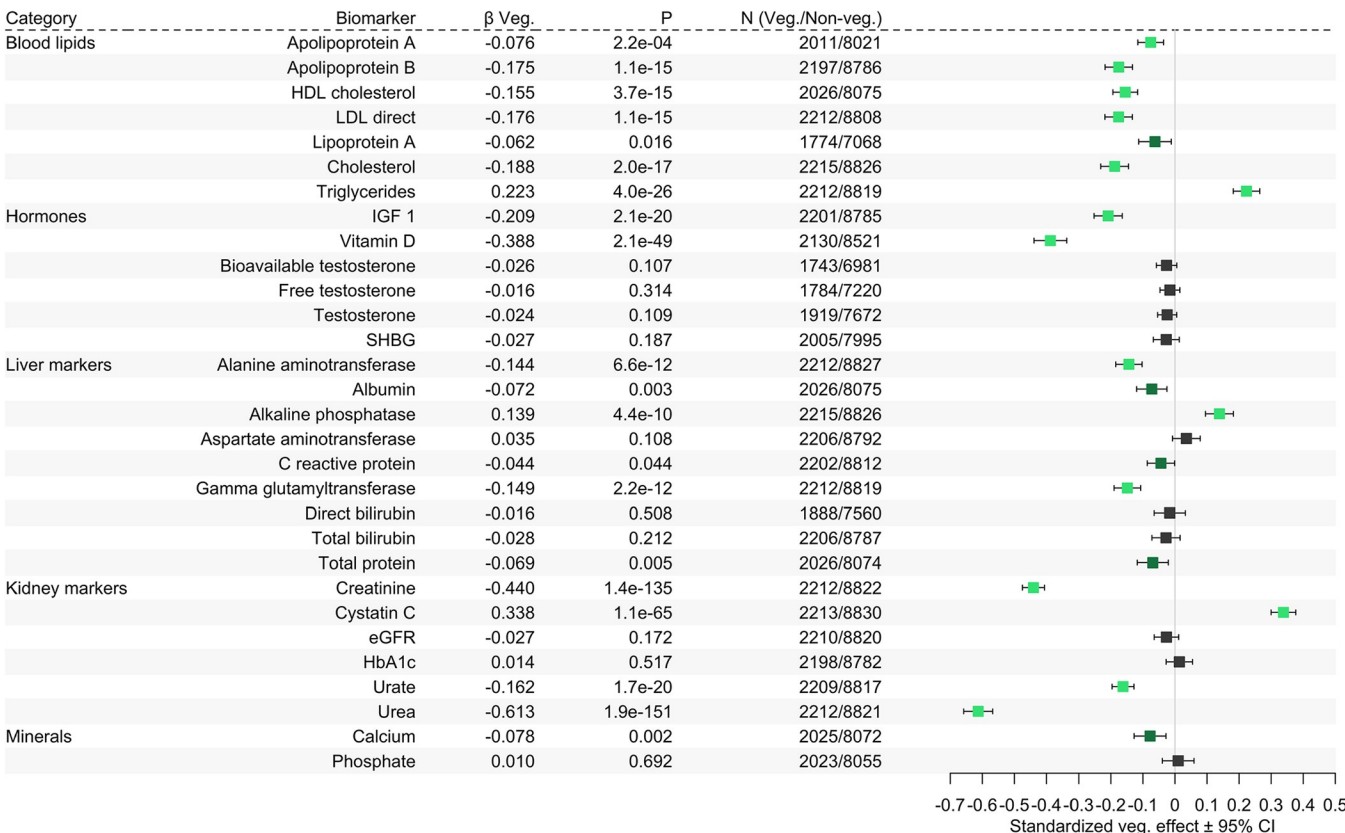

**Fig 2. Forest plot of estimated vegetarianism effects.** Vegetarians were matched 1:4 with nonvegetarians and effects of vegetarianism were estimated across thirty biomarkers. Error bars show 95% confidence intervals. Light green dots indicate multiple-testing-corrected significance ($P<0.0017$), dark green dots show nominal significance ($P<0.05$), and black dots are not significant.

the growth hormone-regulating insulin-like growth factor 1 (IGF-1). Sex-related hormone measures of testosterone (total T, bioavailable-T, and free-T) and sex hormone binding globulin (SHBG) were not significant in the combined nor sex-stratified effects estimation (S4 Fig).

Alanine aminotransferase (ALT) and gamma-glutamyl transferase (GGT) were negatively associated with vegetarianism, while a positive effect was observed with alkaline phosphatase (ALP). Effects on other liver-associated markers such as albumin, aspartate aminotransferase, C-reactive protein (CRP), direct bilirubin, total bilirubin, and total serum protein, were not significant. Kidney markers associated with protein metabolism and breakdown, such as creatinine, urate and urea, displayed negative effects from vegetarianism, while cystatin C was associated with a strong positive effect, and eGFR did not have a significant association. HbA1c (glycated haemoglobin) was not significantly associated. Vegetarianism had a negative effect on serum calcium that nearly reached the multiple testing threshold ($P = 0.002$), while phosphate was not significantly associated. Sex-stratified analysis revealed the effect of vegetarianism was driven by one sex in three traits: ApoA was significant only in males, while ALP and Lp (a) were significant only in females. C-reactive protein was nearly significant ($P = 0.002$) in females (S4 Fig and S4 Table).

## Genome-wide association study of vegetarianism

A total of 7,918,739 variants were tested in a GWAS of 152,764 European UKB participants, using vegetarianism as a binary outcome as defined in Table 1. In standard and BMI-adjusted

models, $P$-values were highly correlated ($R$ = 0.97; S5 Fig). Potential inflation from imbalanced case:control ratio (2,312 vegetarians, 152,764 nonvegetarians) was adjusted for by regenie ($\lambda$ = 1.032 in both models). No associations with vegetarianism reached genome-wide significance (GWS; $P<5\times10^{-8}$; S6A Fig and S5 Table). The two most significant variants in both models were indels, at 4:183448129_AT_A ($P_{standard}$ = 1.65×10$^{-7}$; $P_{adj-BMI}$ = 1.36×10$^{-7}$) and 11:870094_CG_C ($P_{standard}$ = 1.61×10$^{-7}$; $P_{adj-BMI}$ = 2.10×10$^{-7}$). GWAS $P$-values were aggregated into genic regions using MAGMA. No genes achieved statistical significance after multiple testing correction (S6B Fig). The most significant genes in each model were *HLA-DPB1* ($P_{standard}$ = 1.12×10$^{-5}$; $P_{adj-BMI}$ = 5.73×10$^{-5}$) and *YWHAZ* ($P_{standard}$ = 1.44×10$^{-4}$; $P_{adj-BMI}$ = 4.66×10$^{-5}$).

## Genome-wide gene-vegetarianism interactions

**Variant-level interactions.**   Gene-environment interaction analyses using vegetarianism status (Table 1) as the environmental exposure were performed across 30 serum biomarker traits ($N$ = 117,356–147,253) using standard and BMI-adjusted models. We were specifically interested in 1 degree of freedom interaction effects and their corresponding $P$-values, as these most directly demonstrate the effects modification of vegetarianism on genetic associations, or vice versa. Genomic control ($\lambda$) for robust $P$-values ranged from 0.985–1.024, indicating no issues of inflation (S5 Table).

Across the 30 biomarkers analyzed for gene-vegetarianism interactions, only one variant attained GWS, and no variants reached significance at a stricter threshold corrected for the number of traits analyzed ($P< 1.67\times10^{-9}$; S7 Fig and S6 Table). In calcium, interaction of rs72952628 (chr4:146,637,234 C/T; MAF = 0.04) was GWS in the standard model (Table 2) and nearly in the BMI-adjusted model ($P$-int$_{standard}$ = 4.25×10$^{-8}$; $P$-int $_{adj-BMI}$ = 6.29×10$^{-8}$; Fig 3A), while the marginal $P$-value was high ($P$-marginal$_{standard}$ = 0.027; $P$-marginal$_{adj-BMI}$ = 0.023), indicating predominately interaction effects at this locus. This variant is in the intron of *C4orf51*, and in moderate linkage disequilibrium ($r^2$: 0.61–0.72) with variants in exon 7 of *MMAA* (Fig 3B). In a genotype-stratified regression model adjusted for covariates, vegetarianism was associated with a 0.14-unit decrease (95% confidence interval = [-0.180, -0.0897]; $P$ = 5.41×10$^{-9}$) of serum calcium in homozygotes of the major allele (CC), but with a 0.298-unit increase ([0.136, 0.459]; $P$ = 3.08×10$^{-4}$) in heterozygotes (Fig 3C).

Overall, we found 455 unique trait-interaction loci (from 663 interactions in standard and BMI-adjusted models) which reached suggestive significance ($P<1\times10^{-5}$; S7 Table). Sex-

**Table 2. Significant gene-vegetarianism interactions.** Three interactions were reached genome-wide interaction significance: rs72952628 (*MMAA*) in calcium (variant-level testing), plus *RNF168* in testosterone and *DOCK4* in eGFR (gene-level testing). $P$-int, $P$-value of (1 degree of freedom) interaction; CHR:POS, chromosome:position (hg19); eGFR, estimated glomerular filtration rate. *$P$-values correspond to indicated test type. Interaction significance threshold was $P<5\times10^{-8}$ for variant-level analysis and $P<2.75\times10^{-6}$ for gene-level analysis.

| Trait | Test type | $P$-int* | Top SNP | Candidate gene | CHR:POS | Biological relevance of candidate gene | Description of interaction |
|---|---|---|---|---|---|---|---|
| Calcium | Variant | 4.25×10$^{-8}$ | rs72952628 | *MMAA* | 4:146,637,234 | Facilitates calcium-dependent transcobalamin-bound B$_{12}$ uptake into mitochondria; B$_{12}$ is most common vegetarian nutritional deficiency. | Minor allele (T) modifies effect of vegetarianism from calcium-decreasing to calcium-increasing. |
| Testosterone | Gene | 1.45×10$^{-6}$ | rs73219637 | *RNF168* | 3:196,194,654–196,232,639 | Involved in testosterone production, spermatogenesis, and testicular cancer. | Minor allele (C) modifies effect of vegetarianism from testosterone-decreasing to testosterone-increasing. |
| eGFR | Gene | 6.76×10$^{-7}$ | rs17159341 | *DOCK4* | 7:111,844,643–111,984,989 | Associated with kidney markers such as glucose levels, blood pressure, type 2 diabetes, and dehydroepiandrosterone sulphate. | Minor allele (G) modifies effect of vegetarianism from eGFR-increasing to eGFR-decreasing. |

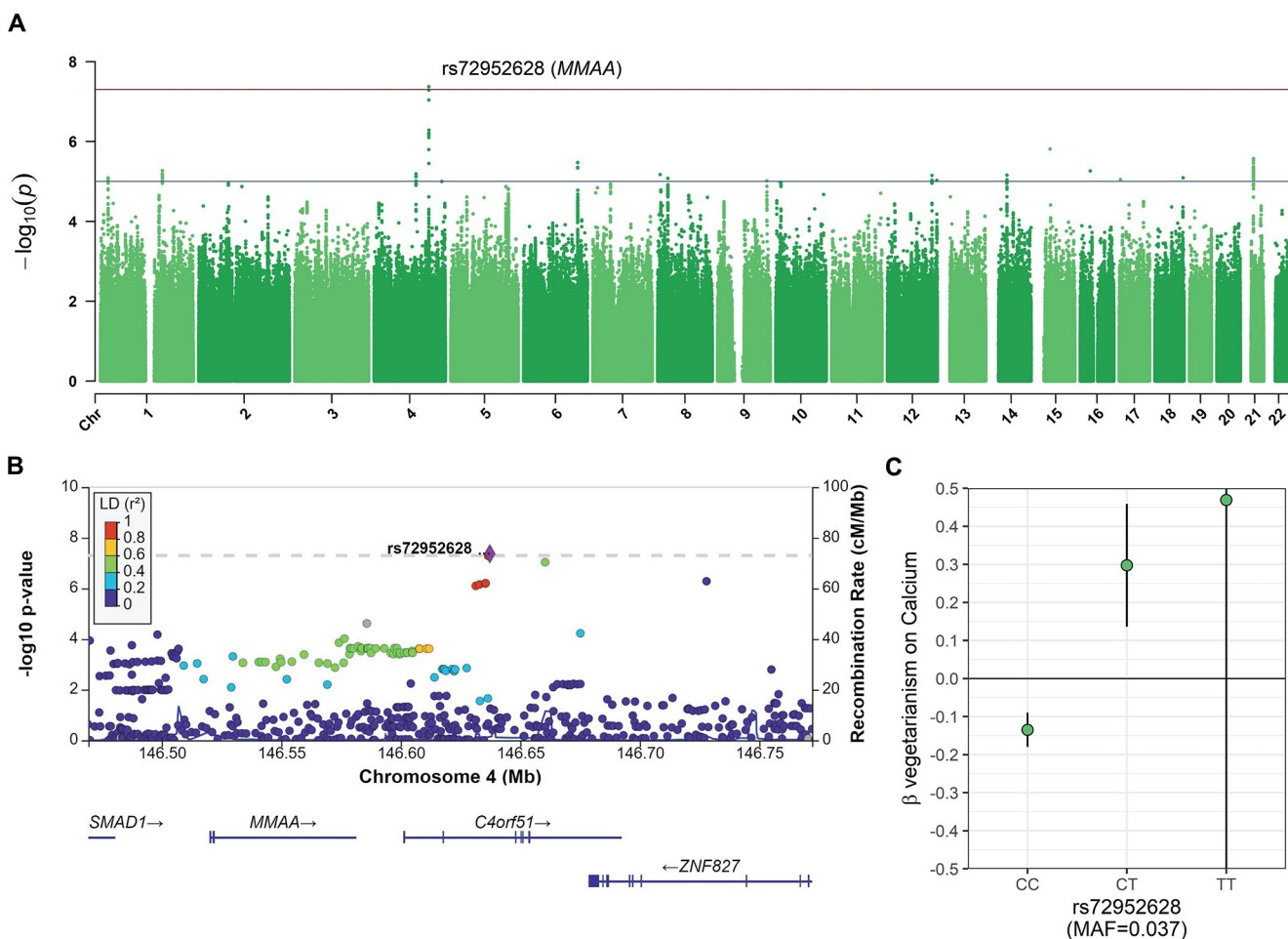

**Fig 3. Calcium gene-vegetarianism interaction at rs72952628. (a)** Manhattan plot of *P*-values for gene-vegetarianism interaction on calcium. One variant, rs72952628 (chr4:146,637,234), passed the genome-wide significance threshold of *P*<5×10⁻⁸ (red line). **(b)** The regional Manhattan plot of rs72952628 shows that it is in linkage disequilibrium with variants in *C4orf51* and *MMAA*. **(c)** Plot stratified by genotype shows that minor allele modifies the effect of vegetarianism on calcium from negative to positive. The homozygous minor allele genotype, TT, has a large error bar because of its low frequency in the sample (n = 207). Error bars show 95% confidence intervals; units of calcium are SD.

stratified models of all interactions reaching suggestive significance across 30 biomarkers did not reveal any additional GWS interactions present in a single sex (S7 Table). In general, females had more significant interaction *P*-values (S8 Fig), likely owing to the larger number of vegetarians (1,462 vs. 758 males)

The locus at *MMAA* has not previously been associated with calcium. MMAA is a GTPase involved in one-carbon metabolism of vitamin B₁₂ (B₁₂; also known as cobalamin). MMAA mediates the transport of cobalamin (Cbl) into mitochondria for the final steps of adenosylcobalamin (AdoCbl) synthesis. The most prominent cause of B₁₂ deficiency is inadequate dietary intake, and this is especially common among vegetarians and vegans because the majority of dietary B₁₂ is derived from animal sources [31]. GTEx single-tissue expression Quantitative Trait Loci (eQTL) analysis for rs72952628 showed an exclusive and significant association with *MMAA* gene expression in four tissue types, where the heterozygote CT was consistently associated with higher *MMAA* expression (S9 Fig). Across the 54 tissues and cells examined by GTEx, *MMAA* had the highest expression in the liver (S10A Fig). With regard to calcium, of the fifteen or more gene products involved in B₁₂ transport and processing [31], two have

calcium-binding domains: cubilin, and CD320. In the distal ileum, binding of the IF-B$_{12}$ complex to the cubilin receptor is calcium-dependent [32]. However in the B$_{12}$ pathway, *MMAA* is more closely related to the CD320 receptor, which mediates Ca$^{2+}$ dependent transcobalamin-bound B$_{12}$ cellular uptake [33].

**Gene-level interactions.** Interaction *P*-values of GWIS variants were aggregated into genic regions using MAGMA for each of the 30 biomarkers. Variants were mapped to 18,208 genes, and the significance threshold corrected for the number of genes was $P<2.75\times10^{-6}$; this threshold additionally corrected for the number of traits was $P<9.15\times10^{-8}$. Genomic control (λ) for these aggregated models ranged from 0.898–1.098 (S11 Fig and S5 Table). Two genes in two traits were significant at the threshold corrected for the number of genes (Table 2): *RNF168* in total testosterone ($P_{\text{standard}} = 1.45\times10^{-6}$, $P_{\text{adj-BMI}} = 1.03\times10^{-6}$; Fig 4A–4C), and *ZNF277* in eGFR ($P_{\text{standard}} = 6.76\times10^{-7}$, $P_{\text{adj-BMI}} = 9.28\times10^{-6}$; Fig 4D–4F). No genes were found to be significant at the more conservative significance level corrected for the number of traits (S6 Table).

*RNF168* had the highest expression levels in the testis in GTEx (S10B Fig). *RNF168* has previously been associated with testosterone levels in UKB GWAS [34]. The top interaction variant at this locus was rs73219637 ($P$-int$_{\text{standard}}$ = $7.13\times10^{-7}$; $P$-int$_{\text{adj-BMI}}$ = $1.02\times10^{-6}$; $P$-marginal$_{\text{standard}}$ = 0.43; $P$-marginal$_{\text{adj-BMI}}$ = 0.34; Fig 4B). A genotype-stratified model indicated vegetarianism is associated with a 0.039-unit decrease in testosterone ([-0.068, -0.010]; $P$ = 0.009) in TT individuals, but with a 0.175-unit increase ([0.100, 0.249]; $P$ = $4.44\times10^{-6}$) in TC individuals (Fig 4C). The heterozygote (TC) is also associated with an increased expression of *RNF168* in the testis ($P$ = $4.81\times10^{-3}$), though this did not pass the GTEx multiple testing significance cutoff (S10B Fig). RNF168 is involved in the repair of DNA double-strand breaks, and mutation of this gene is associated with Riddle syndrome, symptoms of which include increased radiosensitivity, immunodeficiency, motor control and learning difficulties, facial dysmorphism, and short stature. A mouse model of Riddle syndrome found RNF168 deficiency caused decreased spermatogenesis, and *RNF168* was identified as a tumor suppressing candidate gene in testicular embryonal carcinomas [35].

Because vegetarianism is correlated with sex, and testosterone is a highly sex-dependent trait, we performed a sex-stratified and genotype-stratified analysis of the lead interaction variant at *RNF168* (rs73219637) to verify that the observed gene-vegetarianism interaction was not indirectly produced by a gene-sex interaction. In both males and females, differences remained in vegetarianism effect across genotype strata (S12 Fig and S8 Table). This preservation of the interaction effect in sex-stratified analysis suggests that this finding was not a result of confounding by sex differences.

While *ZNF277* contains a number of variants with suggestive interaction *P*-values, the lead variant in this region is rs17159341 ($P$-int$_{\text{standard}}$ = $2.58\times10^{-7}$; $P$-int$_{\text{adj-BMI}}$ = $8.61\times10^{-7}$; $P$-marginal$_{\text{standard}}$ = 0.09; $P$-marginal$_{\text{adj-BMI}}$ = 0.10; Fig 4E), found in the first intron of *DOCK4*. A genotype-stratified model of rs17159341 indicated 0.132-unit increase ([0.089, 0.174]; $P$ = $1.09\times10^{-9}$) of eGFR for those with the AA allele, and 0.416-unit decrease in GG carriers ([-0.658, -0.174]; $P$ = $7.62\times10^{-4}$; n = 3,374), with no significant effect in heterozygotes (Fig 4F).

*DOCK4* ($P_{\text{standard}}$ = $6.05\times10^{-6}$, $P_{\text{adj-BMI}}$ = $2.12\times10^{-5}$) appears to be a more relevant candidate gene than *ZNF277*. Though *DOCK4* has not been directly associated with eGFR in GWAS, it has been associated with several traits related to kidney health, such as diastolic blood pressure, type 2 diabetes, dehydroepiandrosterone sulphate measurement (a marker for adrenal disorders) and "Water consumption (glasses per day)." A recent study demonstrated that *in vivo* and *in vitro* DOCK4 expression was found to increase with high-glucose, and that DOCK4 could reverse USP36-induced epithelial-to-mesenchymal transition effect, which is involved in diabetic renal fibrosis and nephropathy [36].

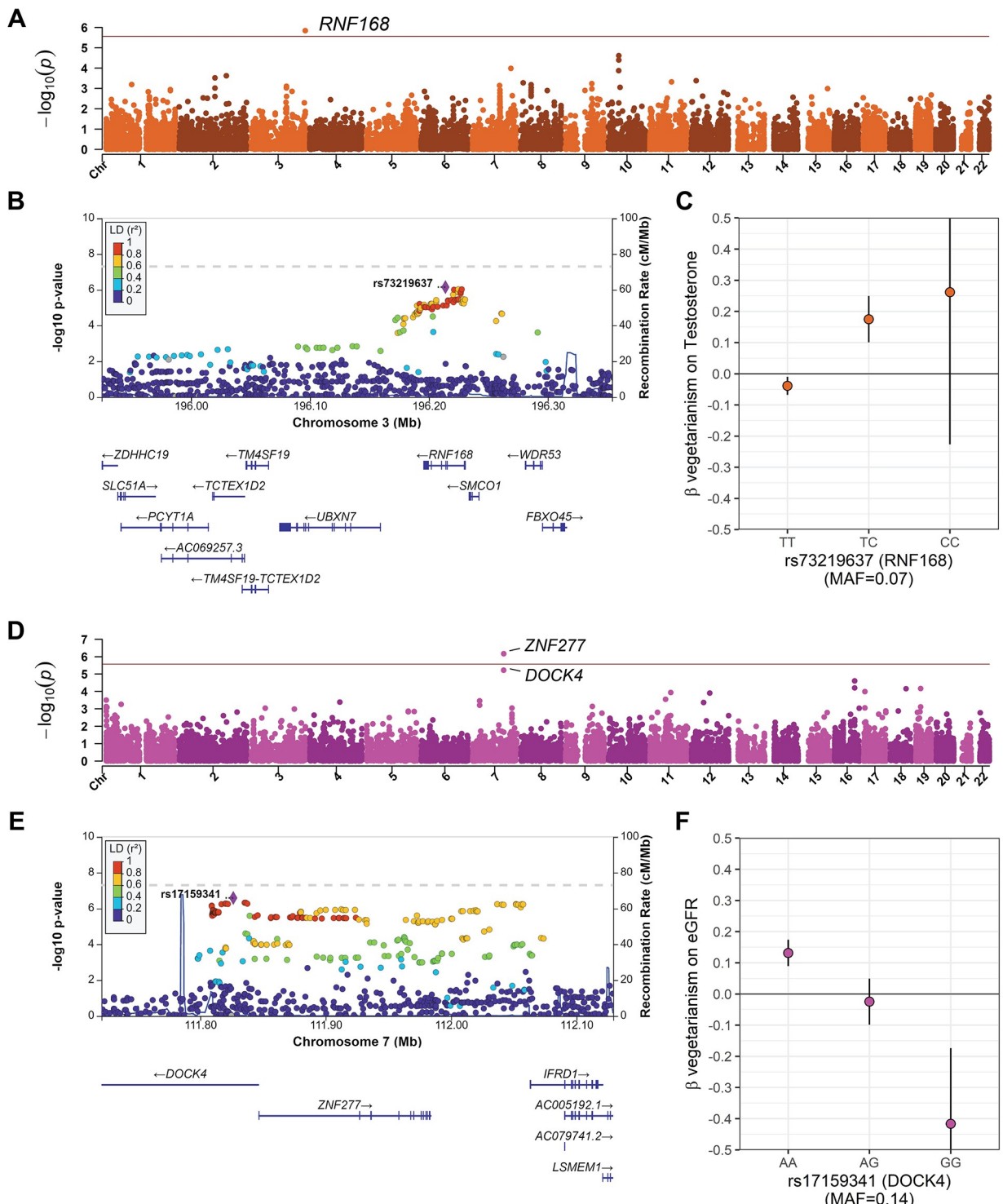

**Fig 4. Gene-level gene-vegetarianism interactions in testosterone and eGFR.** Gene-level Manhattan plots, regional association plots, and genotype-stratified vegetarianism effects for two traits, **(a-c)** testosterone and **(d-f)** estimated glomerular filtration rate (eGFR), which had interactions that reached significance at a level corrected for the number of genes tested (red line at $P = 2.75 \times 10^{-6}$).

**Comparison with single-criterion vegetarianism.** As a sensitivity analysis of the three significant interaction loci, interaction analyses were performed using a single-criterion vegetarianism exposure variable, determined only by indication of "Never" eating meat or fish on the IA. At all three index variants, interaction *P*-values were at least nominally significant (S9 Table). However, despite a larger sample of vegetarians (ranging from 4,626 to 5,373 in the three traits), we observed consistently weaker interaction effects and less significant interaction *P*-values in the single-criterion vegetarianism model than in the strict vegetarianism model (S13 Fig). This suggests that the increase in power derived from our precise definition of vegetarianism was more meaningful in increasing signal strength than the approximately 59% reduction of case size incurred by filtering participants.

## Discussion

In this study we developed a multi-step approach to evaluate the health impacts of vegetarian diets, using both traditional and genetic epidemiological methods, the latter of which has not previously been applied to vegetarianism. Initially, we found several patterns in the UKB participant data that indicated rigorous quality control was necessary to identify strict vegetarians. For example, at each instance of the 24HR, we found that about 10–14% of self-identified vegetarians indicated eating fish, meat, or both, on that same instance of the dietary survey (Fig 1D). Additionally, of 1,229 participants who indicated they "have never eaten meat in [their] lifetime," (IA, Field 3680), 132 (10.7%) also indicated on the same instance of the dietary questionnaire that they occasionally eat oily fish, with 83 of these participants (6.8%) even indicating they eat oily fish once a week or more (S14 Fig). It seems likely that a portion of vegetarians consider eating fish as compatible with the diet, despite this contradicting common parlance. We also observed that in the three-year period of 24HR administration between April 2009 and June 2012, many participants either stopped identifying as vegetarian, or began identifying as such (Fig 1C). Duration of vegetarianism adherence has been shown to impact health measures in several studies (i.e., vegetarianism is a "time-dependent exposure") [8,19,37–39].

The impact of our strict definition of vegetarianism is highlighted in two aspects of our results. When comparing our vegetarianism effects estimates (Fig 2) with a previous UKB study [40], which used only the IA to designate vegetarians and did not use participant matching nor rank-transformed biomarkers, our results and theirs were directionally consistent, with two notable exceptions. SHBG, which we found to have a slightly negative, non-significant effect, was significantly higher in their vegetarian and vegan cohorts; and CRP, which had a weak negative effect in our analysis (*P* = 0.044), was appreciably lower in their vegetarian and vegan cohorts. These differences are also apparent in our plots of raw biomarker values comparing unmatched and matched nonvegetarians (S2 Fig). Next, our three novel interactions were found to be weaker and less significant in our G×E sensitivity analysis using a "noisier" single-criterion vegetarianism as the exposure (S13 Fig and S9 Table). Given that these three interactions were discovered using the strict vegetarianism definition, a decrease in significance might be expected using any other definition, based on a similar argument to the "winner's curse" across samples [41]. However, winner's curse is not sufficient to explain the substantial differences in significance observed between these two exposure definitions. We propose our strict vegetarianism definition was necessary to produce adequate signal strength to identify these three novel and biologically relevant interaction loci. Taken together, these distinctions suggest that vegetarian self-identification should be treated with caution in dietary surveys, and single-criterion designations of vegetarianism can potentially produce noise or spurious associations [38].

We estimated the effects of vegetarianism on 30 serum biomarkers using a model with matched participants that did not consider genetic effects, and found vegetarianism had significant effects on 15 of these biomarker traits (Fig 2). The majority of results from our effects estimation can be explained within the context of the restricted dietary cholesterol, increased dietary fiber, and different amino acid profiles found in the plant-based components of vegetarian diets. For example, vegetarianism had significant negative effects on serum levels of total cholesterol, all lipoproteins (LDL, HDL, Lp (a), ApoA, ApoB), and vitamin D, which is synthesized from cholesterol. Although serum cholesterol is mainly derived from *de novo* synthesis in the liver, our results suggest that intake of animal protein may nonetheless produce a significant difference in serum levels of cholesterol and related molecules. Alternatively, these differences may be attributable to higher levels of fiber in plant-based diets, which reduces cholesterol and overall inflammation [42]. Interestingly, vegetarianism had a significant and moderate triglyceride-raising effect. This adds to previous evidence that a vegetarian diet may actually raise triglycerides [43,44], though recent large meta-analyses had opposite findings [1,8]. The positive effect on triglycerides may be explained as a consequence of lower vitamin D [45], or higher dietary intake of simple carbohydrates [42] in vegetarians. Conversely, without considering genetic differences, we replicated previous findings [46] that vegetarianism does not have significant effects on the cholesterol-derived sterol hormone testosterone (total T, bioavailable-T, and free-T), nor on the testosterone inhibitor SHBG, in both full and sex-stratified models.

Our results did not clearly indicate benefit nor harm of vegetarianism on biomarkers of liver function. For example, we found that vegetarianism had a significant negative effect on ALT and GGT, lower levels of which are associated with healthier liver function. Conversely, we observed a significant positive effect on ALP; increased levels of ALP have been observed in the context of chronic kidney disease (CKD) and vitamin D deficiency. Several studies have shown a decrease in ALP can be achieved by administering activated vitamin D compounds [47].

Improved kidney biomarkers have been associated with increased plant protein intake [42]. Creatinine and urea, byproducts of protein metabolism, were negatively associated with vegetarianism. This can be explained by lower overall protein intake, differences in amino acid composition, or increased fiber intake in vegetarian diets [42]. Vegetarianism also had a significant negative effect on urate (AKA "uric acid"), which can cause gout, kidney stones, and kidney injury in high amounts, but is also a serum antioxidant [48]. Higher consumption of fiber in plant-based diets has been associated with higher eGFR and a lower risk of developing CKD [42]. The higher overall alkalinity of a vegetarian diet also impacts kidney function [42].

The effect of vegetarianism on serum calcium (without consideration of genetics) was small, negative, and marginally significant ($\beta$ = -0.078; $P$ = 0.002). Serum calcium is regulated by calcitriol (1,25-dihydroxycholecalciferol), the active form of vitamin D made in the kidneys. Calcitriol increases serum calcium by increasing the uptake of calcium from the intestines, and may also increase calcium excretion via decreased parathyroid synthesis [49]. Calcium deficiency is a known risk in vegetarian diets, though it can be improved by increased dairy consumption, and is also indirectly dependent on intake of sodium, caffeine, and total protein [50].

We did not find a "vegetarianism gene," nor any variant that was significantly associated with vegetarianism at GWS. Our null finding is similar to a recent GWAS in a Japanese cohort [26]. We nominally replicated the single GWS variant found in a recent GWAS in UKB Europeans by Yaseen et al. (rs72884519) [25], and found a similar effect size ($\beta$ = -0.09; $P$ = 0.002). We attribute the discrepancy in $P$-values between our study and theirs ($P$ = 4.997×10$^{-8}$) to important differences in our respective vegetarianism definitions. Notably, their vegetarianism

cohort included participants that completed only the IA and not the 24HR. There is no data field on the IA to self-identify directly as vegetarian, and we have described above how single time-point survey data can be unreliable [14–16]. Yaseen et al. [25] therefore included 2,566 vegetarians (~48% of their cases) that we did not classify as such. Furthermore, we cannot determine what is meant when their methods refer to "the diet by 24-hour recall questionnaire, which captured ~110K of the respondents," [25] because the 24HR has spanned approximately 211,000 respondents since its release in September 2012 [51].

Variants in *HLA-DPB1*, the most significant hit in our vegetarianism GWAS gene-based test, have been previously associated with cognitive empathy [52], which could potentially be involved with one's decision to become vegetarian. This connection, while interesting, is speculative, and more evidence is necessary. We also found that for all three significant interaction loci, there were no significant genetic associations in our GWAS of vegetarianism, nor significant genetic marginal effects on their respective biomarker traits. This strengthens the evidence for interaction effects driving the signals at these loci.

We performed GWIS across 30 biomarkers, and identified significant gene-vegetarianism interactions in three traits: a variant-level interaction in calcium, and gene-level interactions in testosterone and eGFR. These represent the first gene-vegetarianism interactions identified to date. The interaction in calcium at rs72952628 was found to be significantly associated with expression changes in *MMAA*, a protein in the $B_{12}$ metabolism pathway. $B_{12}$ deficiency is the highest nutritional concern in vegetarians, and dietary intake plays a primary role in $B_{12}$ availability [1,31]. Interestingly, though we did not directly query $B_{12}$ levels, this pathway was implicated in our results. We have suggested *CD320* as a relevant calcium-dependent protein in this pathway; *CD320* serves as the cellular gateway for transcobalamin-bound $B_{12}$ to the cell [33]. Similarly, we have proposed *RNF168* and *DOCK4* as the most likely candidate genes for the gene-level interactions based on gene expression and experimental evidence related to testosterone and eGFR, respectively. More experimental evidence is needed to validate these proposals, as there may be indirect mechanisms also involved in these interactions. Additionally, none of the three traits found to have GDIs showed significant effects of vegetarianism (at $P<0.0017$) in the traditional (non-genetic) epidemiological analysis, emphasizing the importance of interactions in modeling the phenotypic effects of diets.

Our study had several limitations. First, we performed a one-stage analysis (discovery only) without replication. UKB is among the first datasets with dietary data that is sufficiently powered for a GWIS. The benefit in our study of being able to utilize multiple dietary surveys and criteria in defining vegetarians from UKB, also rendered us unable to produce an equally rigorous set of vegetarians for replication. Nonetheless, we consider these discovery-stage results valuable for several reasons. We have demonstrated the noise in a single-criterion definition of vegetarianism in UKB [38], which is relevant in interpreting other vegetarianism studies. Next, our tandem use of strict vegetarians and algorithmic matching in our traditional epidemiological analysis, which simulates the experimental design of a large-scale randomized control trial, established a greater balance between vegetarians and nonvegetarians than has been achieved in previous analyses [1,8,9].

Although we found significant interactions that passed GWS thresholds, they did not reach a threshold further corrected for thirty traits. In this multi-trait GWIS, the multiple testing burden was high, on top of an already strict GWS threshold. GWIS have sample size requirements which require approximately four times more participants to achieve the same power as in a GWAS with comparable effect sizes [53,54]. We suspect that future studies with larger sample sizes will produce additional significant loci. This is supported by several biologically plausible interactions reaching suggestive significance, for example, *BRINP3* in vitamin D ($P = 3.88 \times 10^{-6}$), and *INTU* in SHBG ($P = 3.93 \times 10^{-6}$).

In contrast with increasingly common recommendations that vegetarianism is universally beneficial for all people [4,6,7,55], we found significant biomarker effects corresponding to potentially worse health in vegetarians. Our traditional epidemiological analysis showed a triglyceride-raising effect of vegetarianism; raised triglycerides are a symptom of metabolic syndrome and a risk factor for heart disease and stroke. Lower vitamin D and higher ALP were also observed, both of which have been associated with negative health outcomes as described above. The vegetarianism effects we observed on urate and testosterone have been associated with depression [48,56], and depression has been repeatedly associated with vegetarianism in observational studies [57].

The emerging paradigms of precision medicine and precision nutrition suggest that genetic makeup should help inform optimal disease treatment strategies. We identified three novel gene-vegetarianism interactions in this study and used available functional analyses to put these interactions into plausible biological context. Overall, our findings are most relevant for European ancestry individuals who are in the same age range (40 to 70 years old) as our study cohort; vegetarian and vegan diets for children [58] and pregnant women [59] are associated with serious risks of malnutrition. These results should be interpreted within the context of the specific nutritional needs of an individual. And, as in any genome-wide study, these statistically significant interactions must be externally replicated and verified experimentally. We hope that researchers will utilize our findings and analysis protocols in future studies to conduct replications and meta-analyses, and to inform clinical trials.

## Methods

### Ethics statement

UK Biobank (UKB) approved the use of medical and genetic data under Project ID 48818. UKB received ethical approval from the National Research Ethics Service Committee North West–Haydock (reference ID 11/ NW/0382) and obtained written informed consent from participants. This project using existing UKB data was approved by the Institutional Review Board (IRB) at the University of Georgia. Data analysis was performed on a University of Georgia high performance computing server with strict data protection protocols and two-factor authentication. Participants that withdrew their consent as of Feb. 22nd, 2022 were removed (N = 114).

### Vegetarianism designation

UKB is a prospective cohort study containing >500,000 participants between ages 40 and 70, who were recruited in England, Scotland, and Wales between 2006 and 2010. All UKB Field and Category references can be located in their publicly available data dictionary (https://biobank.ndph.ox.ac.uk/ukb/). Dietary data was collected in two separate surveys. All participants answered the touchscreen questionnaire on "Diet" during their initial visit to the Assessment Centre (Category ID 100052). Additionally, the "Diet by 24-hour recall" section of the "Online follow-up questionnaire" (24HR; Category ID 100090) was administered to a subset of participants on a voluntary basis, during the last phase of the initial assessment (Instance 0; N = 70,689) and subsequently via email, for a total of up to five rounds between April 2009 and June 2012 (N = 210,967 unique participants) [60].

Our goal was to identify a strict subset of participants who were most likely to have consistently followed a vegetarian or vegan diet at the time of the blood draw for biomarker measurement at the IA. Vegetarians and vegans were grouped together as "vegetarians" in all analyses to increase statistical power. In this study vegetarians were defined as meeting all four of the following criteria. First, in a participant's first instance of taking the 24HR, in response to the

question "Do you routinely follow a special diet?" (Field 20086), they must have indicated "Vegetarian diet (no meat, no poultry and no fish)" and/or "Vegan diet". Next, on that same first instance of the 24HR, a participant must have also answered "No" to "Did you eat any meat or poultry yesterday? Think about curry, stir-fry, sandwiches, pie fillings, sausages/burgers, liver, pate or mince," (Field 103000) as well as to "Did you eat any fish or seafood yesterday? e.g. at breakfast, takeaway with chips, smoked fish, fish pate, tuna in sandwiches." (Field 103140). Third, on the initial dietary assessment survey, participants must have answered "Never" to all of the questions asking how often meat or fish was eaten (Fields 1329, 1339, 1349, 1359, 1369, 1379, and 1389). Finally, on the initial assessment, participants must have answered "No" to the question "Have you made any major changes to your diet in the last 5 years?" (Field 1538).

## Phenotype data

Continuous serum biochemistry markers were obtained from UKB Category 17518. Oestradiol and rheumatoid factor (Fields 30800, 30820) were excluded due to limited participant data (<20% of participants). Glucose (Field 30740) was excluded due to inconsistencies in fasting times among participants and a limited number of participants with fasting times larger than 7h. Total cholesterol, LDL-C, and apolipoprotein B were divided by an adjustment factor (0.749, 0.684, and 0.719, respectively) for those who self-reported use of statins [34]. Three derived traits were also included. Free testosterone was calculated with the Vermeulen equation [61], bioavailable testosterone was calculated with the Morris equation [62], and the CKD-EPI Creatinine-Cystatin Equation was used to calculate estimated glomerular filtration rate (eGFR) [63]. All traits were transformed using direct rank-based inverse normal transformation (RINT) with random separation of ties.

## Genotype data

Genotype data was provided with initial quality control and imputation with Haplotype Reference Consortium (HRC) and 1000 Genomes variants by UKB (v3) as previously described [64]. Additionally, we removed variants with imputation quality score (INFO) <0.5, minor allele frequency (MAF) <1%, missing genotype per individual >5%, missing genotype per variant >2%, or Hardy-Weinberg equilibrium (HWE) $P<1\times10^{-6}$. Variant filtering and genotype file format conversions were performed using PLINK2 alpha-v2.3 [65]. After quality control, 7,918,739 variants remained. All genomic positions in this study refer to the Genome Reference Consortium Human Build 37 (GRCh37; hg19).

## Participants

Only participants designated as having European (EUR) ancestry by the Pan UKBB project [30] were used in analyses to avoid population stratification. Participants were removed on the following quality control parameters: mismatches between self-reported and genetic sex, poor quality genotyping as flagged by UKB, sex chromosome aneuploidy, and/or having a high degree of genetic kinship (ten or more third-degree relatives as identified in Field 22021). Additionally, we iteratively identified and removed each individual with the highest number of relationships, and removed the minimum number of participants to eliminate all related pairs.

## Sample matching and estimating vegetarianism effects

To select controls for the analysis of vegetarianism exposure effects, each case was pre-processed to match four controls with nearest-neighbor (greedy) matching without replacement,

using MatchIt v4.4.0.9004 [66]. Matching distance between participants was calculated by general linearized model, and was performed on the basis of age, sex, body mass index (BMI; kg/m$^2$), alcohol use frequency (<3 drinks/week or ≥3 drinks/week), previous smoking status (yes/no), current smoking status (yes/no), standardized Townsend deprivation index, and the first five genetic principal components. Sixteen vegetarians with incomplete covariate information were excluded, leaving a total of 2,312 EUR vegetarians.

Matching was followed by regression using the same vector of covariates; using the same covariates is recommended to reduce the dependence of regression estimates on modeling decisions, increase precision, reduce bias, and increase robustness of the effect estimate [20,67]. Vegetarianism effects estimates were computed by linear model (no interaction) in R v4.2.1 with cluster-robust standard errors implemented by Sandwich v3.0–2 [68]. Sex-stratified models of the same matched participants were also run including BMI as a covariate. Forest-plot v3.0.0 was used to make forest plots. For comparison of our results with Tong et al. [40], we considered the results presented in their main tables (model 3), whose model included covariates most similar to ours.

## Genome-wide association study of vegetarianism

GWAS of vegetarianism was performed using Regenie v3.1.2 [69]. Vegetarianism status as defined above was used as a binary trait. A whole genome regression model was fit at a subset of genetic markers from non-imputed UKB genotype calls. Variants used in model fitting were filtered in PLINK2 alpha-v2.3 [65] by these criteria: MAF<0.01, minor allele count<100, genotype missingness<0.1, HWE exact test $P<10^{-15}$. Covariates used for both model fitting and GWAS (standard model) were age, sex, genotyping array, alcohol use frequency, previous smoker (yes/no), current smoker (yes/no), standardized Townsend deprivation index, and the first ten genetic principal components as provided by UKB. A BMI-adjusted model was separately run as a sensitivity analysis for potential confounding effect of BMI. Individuals with missing phenotype or covariate data were excluded. Firth correction was applied for variants with uncorrected $P<0.01$ to reduce the bias in the maximum-likelihood estimates, using a penalty term from Jeffrey's Prior [70]. Genomic control (λ) was calculated for $P$-values using the median of the chi-squared test statistics divided by the expected median of the chi-squared distribution.

## Genome-wide interactions with vegetarianism

GEM (Gene–Environment interaction analysis in Millions of samples) v1.4.3 [71] was used to perform genome-wide interaction studies (GWIS) of 30 continuous biomarker traits, using vegetarianism status as a binary exposure variable. Covariates used in GWIS were the same as in GWAS above. Individuals with missing phenotype or covariate data were excluded. Robust standard error correction as implemented by GEM was performed in all models to correct for initially observed heteroskedasticity. Initial genomic control (λ) using non-robust standard errors ranged from 0.895–1.255, likely due to heteroskedasticity, therefore robust standard errors as implemented by GEM were used for all models. Interaction effects and $P$-values refer to 1 degree of freedom (1df) tests of interaction. Marginal effects refer to the association between genetic effects and phenotype in a model without interaction. A BMI-adjusted model was separately run for all traits. Correlation between effects and $P$-values of standard and BMI-adjusted models was assessed using a two-sided Spearman's rank correlation coefficient. Genotype-stratified generalized linear models included the same covariates as GWAS. Sex-stratified models were run in GEM for interaction variants reaching suggestive significance ($P<1\times10^{-5}$).

Variants were queried for associations with gene expression levels in tissues using Genotype-Tissue Expression (GTEx) Project (GTEx) Analysis Release V8 (dbGaP Accession phs000424.v8.p2). Fastman v0.1.0 [72] was used to generate Manhattan plots. Hudson (v1.0.0) [73] was used to create interactive Miami plots (online).

### Gene-based interaction analyses

MAGMA v1.10 [74] was used to aggregate *P*-values from individual variant associations (for vegetarianism) and 1 df interactions (for 30 biomarkers) to genic regions. Variants were mapped to a total of 18,208 genes using a window of +2 Kb upstream and -1 Kb downstream of the transcription start and stop sites to allow for the inclusion of nearby regulatory variants. Linkage disequilibrium was estimated using reference data from the 1000 Genomes British population of EUR ancestry. The "multi model" method of aggregation was used to apply both "mean" and "top" models and select the one with the best fit [75].

### Comparison with single-criterion vegetarianism

Gene-vegetarianism interaction analyses at the three significant interaction variants were performed in EUR UKB participants using "single-criterion" vegetarianism, determined only by indication of "Never" eating meat or fish initial assessment (Fields 1329, 1339, 1349, 1359, 1369, 1379, and 1389) as the exposure variable ($N_{vegetarian}$ = 5,646; $N_{nonvegetarian}$ = 351,304). This cohort included the strict vegetarian cohort. All model QC, parameters, and covariates were the same as the strict vegetarianism GWIS as described above.

### Data sharing

Full and annotated code used in this analysis, gene-level summary statistics, and annotated interactive Miami plots are publicly available at https://michaelofrancis.github.io/VegetarianGDI/.

Summary statistics for GWAS and GWIS were deposited in GWAS Catalog (https://www.ebi.ac.uk/gwas/). The corresponding accession numbers can be found in S10 Table.

### Supporting information

**S1 Table. Identifying vegans in UK Biobank.** Of 728 UK Biobank participants who self-identified as vegan at least once on a 24-hour recall (24HR) survey. Vegans were assessed on four criteria: self-identifying as vegan on the first 24HR that they participated in, not eating meat or fish on the first 24HR, not eating meat or fish on the initial assessment (IA), and no major dietary changes over the past 5 years. Only 208 UK Biobank participants met these four criteria (top row, bold). This table shows counts from all UK Biobank participants who took the 24HR (N = 210,967). After filtering by ancestry, the total of 208 was reduced to 133 European vegetarians.
(XLSX)

**S2 Table. Participant characteristics.** Categorical covariates (top), continuous covariates (middle) and biomarker phenotypes (bottom) for 155,375 European UK Biobank participants used in analyses. Continuous variables are represented as: mean (standard deviation). Values are shown for full cohort and cohort stratified by strict vegetarianism (as defined in Table 1).
(XLSX)

**S3 Table. Matching summary.** Top: matchit function call used for 1:4 matching of vegetarian and nonvegetarian participants for use in effects estimation analysis. Middle: Summary of

balance for all data and for matched data shows the results of matching on relevant lifestyle factors and genetic principal components one through five. Bottom: Sample sizes in control (nonvegetarian) and treated (vegetarian) samples before and after matching. Std. Mean Diff., standardized mean difference; Var. Ratio, variance ratio; eCDF Mean, mean empirical cumulative density functions to assess imbalance across entire covariate distribution; eCDF Max, maximum eCDF difference, also known as the Kolmogorov-Smirnov statistic.
(XLSX)

**S4 Table. Vegetarianism effects on biomarkers.** Effects of vegetarianism across 30 traits in (left) full and (right) sex-stratified matched groups. BetaVeg, effect of vegetarianism; SE, standard error; M, males only; F, females only. Green P-values are those which were significant after multiple testing correction ($\alpha = 0.05/30$).
(XLSX)

**S5 Table. Summarize GWAS and GWIS.** Quality metrics of genome-wide association with vegetarianism (top) and genome-wide interaction with vegetarianism across 30 biomarker traits (bottom), for variant-level and gene-level analyses. GC $\lambda$, genomic control (lambda).
(XLSX)

**S6 Table. GWAS/GWIS top results.** Most significant results for genome-wide association with vegetarianism (top) and genome-wide interaction with vegetarianism across 30 biomarker traits (bottom), for variant-level and gene-level analyses. All traits analyzed in standard and BMI-adjusted models. Start and stop coordinates of genes represent the +2 Kbp upstream and -1 Kbp downstream window of variant *P*-value aggregation used in MAGMA. CHR, chromosome; POS, position; Nveg, number of vegetarians in analysis; Nnonveg, number of nonvegetarians in analysis; A0, other allele; A1, effect allele; A1Freq, effect allele frequency; Beta Marginal, coefficient estimate for the marginal genetic effect (i.e., from a model with no interaction terms); Beta Marginal SE, robust standard error for beta marginal; Beta G, coefficient estimate for the genetic main effect; Beta G SE, robust standard error for beta G; Beta interaction, coefficient estimate for the interaction term; Beta interaction SE, robust standard error for beta interaction; P marginal, marginal genetic effect *P*-value from robust SE; P interaction, interaction *P*-value from robust SE; P joint, joint test *P*-value (2 degrees of freedom test of genetic and interaction effect) from robust SE. NSNPS, number of SNPs annotated to the top gene; NPARAM, number of relevant parameters used in model; P interaction (MULTI), gene *P*-value for best fit of "mean" and "top" models as determined by MAGMA.
(XLSX)

**S7 Table. Genome-wide gene-vegetarianism interactions reaching suggestive significance.** Full cohort and sex-stratified results for all interactions reaching the suggestive significance threshold of *P*<1e-05. CHR, chromosome; POS, position; Nveg, number of vegetarians in analysis; Nnonveg, number of nonvegetarians in analysis; A0, other allele; A1, effect allele; A1Freq, effect allele frequency; Beta Marginal, coefficient estimate for the marginal genetic effect (i.e., from a model with no interaction terms); Beta Marginal SE, robust standard error for beta marginal; Beta G, coefficient estimate for the genetic main effect; Beta G SE, robust standard error for beta G; Beta interaction, coefficient estimate for the interaction term; Beta interaction SE, robust standard error for beta interaction; P marginal, marginal genetic effect *P*-value from robust SE; P interaction, interaction *P*-value from robust SE; P joint, joint test *P*-value (2 degrees of freedom test of genetic and interaction effect) from robust SE.
(XLSX)

**S8 Table. Genotype-stratified sex-stratified regression models for rs73219637.** Effect estimates of vegetarianism on testosterone in the combined cohort, males only and females only, for each genotype at the top associated variant in *RNF168* in the testosterone gene-vegetarianism interaction analysis. N, number of participants with genotype in each group; SE, standard error; lower, lower 95% confidence interval; upper, upper 95% confidence interval; P, *P*-value of vegetarianism effect.
(XLSX)

**S9 Table. Comparison of strict versus single-criterion vegetarianism interactions.** For each of the three lead variants in vegetarianism interactions, the strict vegetarianism selection we developed produced stronger interaction effects and more significant P-values than a single-criterion vegetarianism definition based only on one survey question. Results for standard and BMI-adjusted models are shown. All models (single and strict crierion) used robust standard errors to correct for heteroskedasticity. CHR, chromosome; POS, position; Nveg, number of vegetarians in interaction analysis; Nnonveg, number of nonvegetarians in the interaction analysis; EA, effect allele; NEA, non-effect allele; EA Freq, frequency of effect allele; G, genotype; SE, standard error.
(XLSX)

**S10 Table. GWAS catalog accessions.** GWAS Catalog accession codes for all variant-level GWAS and GWIS summary statistics generated in this study.
(XLSX)

**S1 Fig. Participant flow chart.** Starting from the entire UK Biobank, visualizing the the number of participants who were excluded in each quality control step, to identify the 2,328 strict European (EUR) vegetarians used in this study. 24HR, 24-hour recall survey, QC, quality control.
(PDF)

**S2 Fig. Boxplots of unadjusted biomarker levels.** Comparing raw values of European strict vegetarians and nonvegetarians across 30 biomarker traits. Boxplots show first decile, first quartile, median, third quartile, and last decile. Units of each biomarker can be found in S2 Table. (**a**) Combined male and female cohort. Dot and label refer to mean. (**b**) Stratified by sex.
(PDF)

**S3 Fig. Love plot of covariates before and after matching.** Plot shows the absolute standardized mean difference of model covariates in nonvegetarians before and after matching with vegetarians for effects estimation. After matching, the ASMD in all model covariates were <0.05 standardized units. BMI, body mass index; AlcoholFreq, frequency of alcohol usage (<3 drinks/week or ≥ 3 drinks/week); zTownsend, standardized Townsend deprivation index; PCA, genetic principal component; distance, matching distance between partici-pants as calculated by general linearized model.
(PDF)

**S4 Fig. Sex-stratified forest plot.** Effects estimation for vegetarianism in sex-stratified BMI-adjusted model. Error bars indicate 95% confidence interval. Bonferroni-corrected significance threshold at $\alpha = 0.05/30 = 0.0017$.
(PDF)

**S5 Fig. Correlation plot comparing P-values of BMI-adjusted GWAS model.** Scatterplot of vegetarian behavior GWAS -log10(*P*) comparing BMI-adjusted versus. BMI-unadjusted

("standard") models. Each point represents one variant. Spearman's Rho (R) and correlation *p*-value shown.
(PDF)

**S6 Fig. Vegetarianism genome-wide association Manhattan and QQ plots.** Manhattan plots and quantile-quantile (QQ) plots showing -log10(*P*) of genetic effects with vegetarianism behavior. Genomic control (λ) for each model is shown in QQ plots. Plots correspond to: **(a)** Variant-level GWAS and **(b)** gene-level GWAS, where *P*-values were aggregated by MAGMA. Top variants with $P<1\times10^{-6}$ are annotated. Top genes ($P<1\times10^{-4}$) in a 5 Mb window were annotated. No variants or genes for vegetarianism behavior were genome-wide significant. All plots shown with and without inclusion of BMI as a covariate.
(PDF)

**S7 Fig. Variant-level gene-vegetarianism interaction Manhattan plots.** Manhattan plots and QQ plots showing the variant-level −log10(P) of genome-wide gene-vegetarianism interaction effects in thirty serum biomarker traits. The blue line corresponds to the genome-wide suggestive threshold (P<1×10–5). In the standard interaction model **(a)**, one trait, calcium, had a significant variant above the genome-wide significance threshold (P<5×10–8; red line). **(b)** No variants were significant in the BMI-adjusted model.
(PDF)

**S8 Fig. Summary of interaction *P*-values in sex-stratified analysis.** Bars show bin counts of interaction *P*-values from males and females for all interactions reaching suggestive significance in the full cohort genome-wide interaction analysis. Sex-stratified models were also run with and without BMI adjustment, as in the full cohort.
(PDF)

**S9 Fig. eQTLs for MMAA and rs72952628.** Violin plots showing expression quantitative trait loci (eQTLs) in four tissues which were significant at the GTEx multiple testing threshold (adipose: subcutaneous, colon: sigmoid, muscle: skeletal, and cells: cultured fibroblasts) plus liver tissue, which nearly reached significance. In all five of these tissues, the heterozygote (CT) shows higher median normalized expression.
(PDF)

**S10 Fig. Bulk tissue gene expression for interaction genes.** Candidate genes for significant interactions with vegetarianism in variant-level and gene-level analyses. Transcripts per million (TPM) shown in tissues ranked from low to high for the genes **(a)** *MMAA*, **(b)** *RNF168*, and **(c)** *DOCK5*.
(PDF)

**S11 Fig. Gene-level gene-vegetarianism interaction Manhattan plots.** Manhattan plots and quantile-quantile (QQ) plots showing the gene-level −log10(*P*) of genome-wide gene-vegetarianism interaction effects in thirty serum biomarker traits. The red line corresponds to the genome-wide significance threshold ($P<2.75\times10^{-6}$; red line). In the standard interaction model **(a)** two traits, estimated glomerular filtration rate (eGFR) and testosterone, had a significant gene. **(b)** Testosterone had one significant gene in the BMI-adjusted model.
(PDF)

**S12 Fig. Genotype-stratified sex-stratified regression models for rs73219637.** Plots show effect of vegetarianism on testosterone in males (left) and females (right), for each genotype at the top associated variant in *RNF168* in the testosterone gene-vegetarianism interaction

analysis.
(PDF)

**S13 Fig. Forest plot comparing strict versus single-criterion vegetarianism interactions.**
For each of the three lead variants in vegetarianism interactions, the strict vegetarianism selection we developed produced stronger interaction effects and more significant *P*-values than a single-criterion vegetarianism definition based only on one survey question.
(PDF)

**S14 Fig. Fish eating frequency of those who have "never eaten meat" in their lifetime.** Bar plot shows non-oily-fish- and oily-fish-eating frequency, reported at the initial assessment, for those who reported on that same dietary survey that they had "never eaten meat in [their] lifetime" (N = 1,230).
(PDF)

## Acknowledgments

This research has been conducted using the UK Biobank Resource under Application Number 48818.Special thanks to the Georgia Advanced Computing Resource Center (GACRC) at the University of Georgia for supporting our data analyses.

## Author Contributions

**Conceptualization:** Michael Francis, Kaixiong Ye.

**Data curation:** Michael Francis.

**Formal analysis:** Michael Francis.

**Funding acquisition:** Michael Francis, Kaixiong Ye.

**Investigation:** Michael Francis, Kenneth E. Westerman, Alisa K. Manning.

**Methodology:** Michael Francis.

**Project administration:** Kaixiong Ye.

**Resources:** Kaixiong Ye.

**Software:** Michael Francis.

**Supervision:** Kaixiong Ye.

**Visualization:** Michael Francis.

**Writing – original draft:** Michael Francis.

**Writing – review & editing:** Michael Francis, Kenneth E. Westerman, Alisa K. Manning, Kaixiong Ye.

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
