## [Decision Letter · Decision Letter 0]

22 Dec 2023

Dear Dr. Ye,

Thank you very much for submitting your manuscript "Gene-vegetarianism interactions in calcium, estimated glomerular filtration rate, and testosterone identified in genome-wide analysis across 30 biomarkers" (PGENETICS-D-23-01075) for review by PLOS Genetics. 

The manuscript was fully evaluated at the editorial level and by independent peer reviewers. As you will see, the reviewers identified a number of serious concerns. Based on the reviews and our editorial evaluation, we regret to say that we will be unable to consider the manuscript further at PLOS Genetics.

The authors’ lack of reference to a similar UKBB-based study by Yaseen et al., published in PLOS One in 2023 and available on medRxiv since October 2022, was a significant concern. In addition, in the absence of replication of any of the findings reported by Yaseen et al., along with some lack of clarity in phenotypes’ definitions, we concluded that the manuscript is unsuitable for publication in PLOS Genetics.

The reviews are attached below this email, and we hope you will find them helpful if you decide to revise the manuscript for submission elsewhere. We are sorry that we cannot be more positive on this occasion.

While we cannot consider your manuscript further for publication in PLOS Genetics, we would like to offer you the option to transfer your submission, with reviews, to PLOS ONE https://www.editorialmanager.com/PONE/

If you DO wish to transfer your submission, please click this link:

<DeepLinkData><DeepLinkTypeID>27</DeepLinkTypeID><peopleID>1047808</peopleID><userSecurityID>1d462be5-75ea-48c4-afed-5ae86d975492</userSecurityID><documentID>41481</documentID><revision>0</revision><manuscriptNumber>PGENETICS-D-23-01075</manuscriptNumber><docSecurityID>56b6adc6-a8d4-484f-9277-4b8fe70a0766</docSecurityID></DeepLinkData>

If you do NOT wish to transfer your submission, please click this link to decline:

<DeepLinkData><DeepLinkTypeID>28</DeepLinkTypeID><peopleID>1047808</peopleID><userSecurityID>1d462be5-75ea-48c4-afed-5ae86d975492</userSecurityID><documentID>41481</documentID><revision>0</revision><manuscriptNumber>PGENETICS-D-23-01075</manuscriptNumber><docSecurityID>56b6adc6-a8d4-484f-9277-4b8fe70a0766</docSecurityID></DeepLinkData>

Please note, all PLOS journals are editorially independent and vary in submission requirements.

Should you choose to transfer, your manuscript files, along with the reviewers' comments and their identities will be transferred automatically, and you will receive a confirmation email within 24 hours. Once transferred, your submission will be returned to you so you can check over your record before completing the submission. You may be asked to provide additional information, such as a response to the reviewers' comments. If you have any questions, please contact the editorial office of PLOS ONE https://www.editorialmanager.com/PONE/

We are sorry that the news is not more positive on this occasion and hope that you will consider PLOS Genetics for other submissions in the future. Thank you for your support of PLOS and of open-access publishing.

Sincerely,

Giorgio Sirugo

Academic Editor

PLOS Genetics

Hua Tang

Section Editor

PLOS Genetics

Reviewer's Responses to Questions

**Comments to the Authors: **

Reviewer #1: The authors present a GWAS on vegetarians vs non-vegetarians which they then use as a binary trait in a genome-wide interaction study for a 30 trait panel of clinical analytes relevant to human health. The paper is on the whole very well written, and was a pleasure to review. The group make excellent use of the UK BioBank dataset, including a very thoughtful and compelling use of the lifestyle questionnaire data which the leverage to curate a retrospective strict vegetarian cohort (in so far as the available data permit this). It is highly unusual for the cohort assembly and sample matching portion of any paper to be particularly compelling, but in this the rigorous whittling of the vegetarian cohort based on inconsistencies in the dietary questionnaire responses made for interesting reading.

Perhaps reflecting the diversity of reasons for the choice of vegetarianism as a lifestyle, and likely in part due to the rigor of the vegetarian cohort curation which resulted in a small cohort size, it is not surprising that no variants met the significance threshold for vegetarianism itself as a GWAS trait. Similarly, GWIS adjusted p-values were generally modest, approaching GWIS-adjusted statistical significance (passing genomewide-significance for individual traits but not when corrected for the 30 traits considered).

The authors highlight selected GWIS associations, but do not overstate their findings nor the implications. Interestingly, multiple GWIS variants map to the vitamin B12 pathway lending genetic support to the long-standing observation of B12 deficiency in strict vegetarians.

Interestingly, the a separate finding from the epidemiological component of this study calls into question the prevailing wisdom that vegetarianism is a universally beneficial lifestyle choice, with the authors noting higher plasma triglycerides and alkaline phosphatase and lower vitamin D, as well as replicating effects on urate and testosterone (the latter very modest).

While the implications of the study are somewhat limited (as the authors note), the study is very well designed, the analysis performed well, and the conclusions are appropriate to the data. This study is very appropriate for publication in PLOS Genetics. The authors should be commended for their inclusion of interactive web versions of their Miami plots.

Minor issues: The phrase "nearly significant" (p16/line 324) is not informative. It would be better if the sentence were reworded, and the non-significant adjusted p-val simply reported instead.

The aesthetics in Fig 1 are odd, and distracting. Why is "first time taking" heavily outlined in Fig1b, and what is the point of the color gradient across the columns in Fig1c? Both are distracting and unnecessary.

The gendered color differences (specifically Fig S2b, and also Fig S4) are unnecessarily subtle (to the point that I didn't initially realize that there was a different fill color for the box plots in Fig S2b). The authors are encouraged to increase the color saturation and/or opacity to make differences easier to distinguish. In fact, it would be more useful to group the data in S2b by sex rather than by cohort (i.e. rather than showing 3 cohorts each split by sex, show 2 sexes split by cohort - the male and female could then be displayed on axes ranged appropriate to the values by sex).

An expanded version of Table 2 would be a useful addition to the extensive supplemental data - perhaps highlighting the top ~5 variants for each trait and the adjusted p-val and beta for each in the cohort as a whole as well as stratified by sex.

Reviewer #2: In this manuscript, Francis and colleagues have scrutinized UK Biobank participants, especially of European ethnicity that practice strict vegetarian diet for genome-wide gene-vegetarianism interaction across 30 metabolic biomarkers. Their gene-diet interaction analysis identified notable gene-vegetarianism interactions on calcium, estimated glomerular filtration rate and testosterone. As authors stated in the manuscript, genetic studies in the context of vegetarianism are limited. As the term "vegetarianism" refers to plant-based varying dietary habits, genetic associations especially with regard to health risk is delicate. It would be logical to explore the genetics of individuals that are strict vegetarians for several generations rather than those choose and practice as a lifestyle. 

A newly published GWAS by Yaseen et al. (2023) identified genetic loci linked to vegetarian diet among white caucasians participated in the UK Biobank study. As the authors have used the same dataset, I am wondering that they missed this GWAS study. It is surprising that two independent GWAS studies performed on the UK Biobank's European vegetarians didn't replicate. I suggest authors to compare their results from the third part (i.e., GWAS to search for variants that may explain vegetarianism behavior on a genetic level) with the significant hits from Yaseen et al. 

1) In page 4 line 77, the following statement "...though a recent GWAS of vegetarianism found none" should be rephrased by citing Yaseen et al. 2023.

Yaseen NR, Barnes CLK, Sun L, Takeda A, Rice JP. Genetics of vegetarianism: A genome-wide association study. PLoS One. 2023 Oct 4;18(10):e0291305. doi: 10.1371/journal.pone.0291305. PMID: 37792698; PMCID: PMC10550162.

2) In page 5 line 84, correct the word "recommandations" to recommendations.

Reviewer #3: In this manuscript the authors describe a multistep study that probes the impact of vegetarianism and vegetarianism-gene interactions on clinical biomarkers in UK BioBank Participants. First, they used diverse convergent evidence to identify strict vegetarians in a manner that is more rigorous than many prior studies. Then they identified lower Vitamin D and cholesterol levels, but higher triglyceride levels in these strict vegetarians. In their first GWAS, no variants associated with vegetarianism (i.e,. no variants predicted the adoption of vegetarianism). In their second set of GWAS on the biomarkers, they identified a genome-wide significant variant-vegetarianism interaction with rs72952628 . Interestingly vegetarians with this variant had higher serum calcium levels. This variant is linked to changes in MMAA expression, a protein involved with B12 metabolism. Many vegetarians have a hard time getting enough B12 and some of its cellular transport is calcium dependent. On the gene level two significant interactions were identified: RNF168 with testosterone, and DOCK4 with estimated glomerular filtration rate (eGFR). Vegetarians with rs73219637 in RNF168 had higher testosterone, and the effect was present in both sexes. Vegetarians who were homozygous for rs17159341 in DOCK4 had lower eGFR. Overall, the manuscript is well written and if these results are corroborated in future studies, they may provide important early evidence for developing precision nutrition care. A few issues deserve consideration:

1) The authors should be applauded for using a multistep process to hone the definition and identification of strict vegetarians. The measurement of diet/environment is usually the most difficult and error prone part of exposure assessment in G-by-E analyses. However even with this robust approach, the vegetarians are not a homogenous group. Could the authors expand their discussion of this heterogeneity and possibly consider subgroup analyses on: 1) vegans and 2) vegetarians who eat fish? 

2) Along these lines, there were 964 people who only partially met vegetarian criteria and it appears that they were included as non-vegetarians in this analysis. It would be useful to see analyses with these participants excluded, rather than reclassified as nonvegetarians. This group may represent intermediate exposure to vegetarianism (or just a subpopulation for whom exposure misclassification is likely). 

3) Line 163 Exactly how were participants eliminated to remove relatives? Was there a pihat threshold? If so, was one of each pair removed in a random selection?

4) The authors should compare and contrast their work with that of Yaseen et al 2023 https://pubmed.ncbi.nlm.nih.gov/37792698/ This study was less comprehensive than the current study, but its findings in a more heterogenous set of UK biobank vegetarians deserve discussion.

5) Brief additional discussion of the biomarkers themselves is warranted. For example: Creatinine is a component of meat and serum creatinine rises when you eat it. This lowers eGFR. In short, the exposure (not eating meat-creatinine) is a component of the outcome (serum creatinine based eGFR). This is an important issue for interpretation, as this way of elevating eGFR is almost tautological and it does not reflect kidney function. Thus this particular outcome may be a little problematic. As for calcium and testosterone levels, it is unclear what might be good/desirable for any given person. Usually higher or lower is not a universal good and there may be sweet spots that differ between subgroups of participants. Mentioning these complexities should help the reader.

**Have all data underlying the figures and results presented in the manuscript been provided?**

Reviewer #1: Yes

Reviewer #2: Yes

Reviewer #3: None

PLOS authors have the option to publish the peer review history of their article (what does this mean?). If published, this will include your full peer review and any attached files.

Reviewer #1: No

Reviewer #2: No

Reviewer #3: No

---

## [Decision Letter · Decision Letter 1]

3 May 2024

Dear Dr Ye,

We are pleased to inform you that your manuscript entitled "Gene-vegetarianism interactions in calcium, estimated glomerular filtration rate, and testosterone identified in genome-wide analysis across 30 biomarkers" has been editorially accepted for publication in PLOS Genetics. Congratulations!

Yours sincerely,

Giorgio Sirugo

Section Editor

PLOS Genetics

Hua Tang

Section Editor

PLOS Genetics

Comments from the reviewers (if applicable):

Reviewer's Responses to Questions

**Comments to the Authors:**

Reviewer #1: The authors have adequately addressed my previous [minor] concerns, and more importantly, have now directly addressed the similar study from Yaseen et al., which was raised by the other two reviewers. Their explanation for the partial overlap in the fiindings between the two studies as reflecting differences in the way the cohorts were subset is plausible and compelling (with the current study excluding ~48% of individuals that met the Yaseen study's "vegetarian" definition).

Reviewer #3: Thank you to the authors for their very thoughtful responses to my comments.

**Have all data underlying the figures and results presented in the manuscript been provided?**

Reviewer #1: Yes

Reviewer #3: None

PLOS authors have the option to publish the peer review history of their article (what does this mean?). If published, this will include your full peer review and any attached files.

Reviewer #1: No

Reviewer #3: No

**Data Deposition**

http://datadryad.org/submit?journalID=pgenetics&manu=PGENETICS-D-23-01075R1

**Press Queries**

---

## [Editor Report · Acceptance letter]

18 Jun 2024

PGENETICS-D-23-01075R1 

Gene-vegetarianism interactions in calcium, estimated glomerular filtration rate, and testosterone identified in genome-wide analysis across 30 biomarkers 

Dear Dr Ye, 

We are pleased to inform you that your manuscript entitled "Gene-vegetarianism interactions in calcium, estimated glomerular filtration rate, and testosterone identified in genome-wide analysis across 30 biomarkers" has been formally accepted for publication in PLOS Genetics! Your manuscript is now with our production department and you will be notified of the publication date in due course.

With kind regards,

Judit Kozma

PLOS Genetics

On behalf of:
